# Phosphorylation of Ykt6 SNARE Domain Regulates Its Membrane Recruitment and Activity

**DOI:** 10.3390/biom10111560

**Published:** 2020-11-16

**Authors:** Pradhipa Karuna M, Leonie Witte, Karen Linnemannstoens, Dolma Choezom, Adi Danieli-Mackay, Mona Honemann-Capito, Julia Christina Gross

**Affiliations:** 1Hematology and Oncology, University Medical Center Goettingen, 37077 Goettingen, Germany; pradhipa.karuna-m@med.uni-goettingen.de (P.K.M.); leonie.witte@med.uni-goettingen.de (L.W.); Karen.Linnemannstoens@zentr.uni-goettingen.de (K.L.); dolma.choezom@med.uni-goettingen.de (D.C.); adi.danieli@stud.uni-goettingen.de (A.D.-M.); mhonema@gwdg.de (M.H.-C.); 2Developmental Biochemistry, University Medical Center Goettingen, 37077 Goettingen, Germany; 3HMU Health and Medical University Potsdam, 14471 Potsdam, Germany

**Keywords:** Ykt6 conformational switch, membrane attachment, protein trafficking, secretory pathway

## Abstract

Sensitive factor attachment protein receptors (SNARE) proteins are important mediators of protein trafficking that regulate the membrane fusion of specific vesicle populations and their target organelles. The SNARE protein Ykt6 lacks a transmembrane domain and attaches to different organelle membranes. Mechanistically, Ykt6 activity is thought to be regulated by a conformational change from a closed cytosolic form to an open membrane-bound form, yet the mechanism that regulates this transition is unknown. We identified phosphorylation sites in the SNARE domain of Ykt6 that mediate Ykt6 membrane recruitment and are essential for cellular growth. Using proximity-dependent labeling and membrane fractionation, we found that phosphorylation regulates Ykt6 conversion from a closed to an open conformation. This conformational switch recruits Ykt6 to several organelle membranes, where it functionally regulates the trafficking of Wnt proteins and extracellular vesicle secretion in a concentration-dependent manner. We propose that phosphorylation of its SNARE domain leads to a conformational switch from a cytosolic, auto-inhibited Ykt6 to an active SNARE at different membranes.

## 1. Introduction

N-ethylmaleimide-sensitive factor attachment protein receptors (SNARE) family members drive membrane fusion by the formation of a trans-SNARE complex consisting of specific v- and t-SNAREs present at vesicle (v) and target (t) membranes. Most SNAREs have autonomously folding N-terminal domains, along with SNARE (coiled-coil) motifs and membrane anchors. The SNARE motifs are 60–70 amino acid residues long [1] and contain repeated heptad patterns of hydrophobic residues. They assemble into parallel four-helix bundles stabilized by a hydrophobic helix that faces the bundle’s core. Within the hydrophobic core of the bundle X-ray crystallography [2,3] revealed an unusual central hydrophilic layer composed of three glutamines (Q) and one arginine (R) residue, which led to the classification of Q- and R-SNAREs, respectively [4].

Ykt6 is an unusual SNARE as it lacks a transmembrane domain and therefore can cycle between cytosol and membranes. Membrane localization depends on the intramolecular interaction of the N-terminal Longin and C-terminal SNARE domains and the presence of a farnesylation and reversible palmitoylation within a CCAIM/CAAX motif at the C-terminus [5]. This interaction is exemplified in yeast, where the release of Ykt6 from endosomal membranes into the cytoplasm depends on a functional Longin domain and an intramolecular interaction with its SNARE domain to fold into a soluble, closed conformation [6]. Additionally, a recent study identified a new geranylgeranyl transferase that plays an essential role for membrane-anchored Ykt6 in proper Golgi function [7].

Albeit sometimes redundant, specific sets of SNAREs mediate distinct steps in intracellular protein trafficking [8]. Correspondingly, Ykt6 interacts with different SNARE partners in vitro [9] and is proposed to function as a membrane stress sensor within the secretory pathway of yeast [10]. In addition to its functions in the homotypic fusion of Endoplasmic Reticulum (ER) and vacuolar membranes [11,12], ER-Golgi trafficking [13] and retrograde Golgi trafficking in yeast [14,15,16], it was also described to function in in autophagy [16] and lysosomal stress [17]. Recent studies revealed that Ykt6 mediates several steps of autophagosome formation in human cells [18], the *Drosophila* fat body [19], and yeast [15,20]. Nevertheless, how Ykt6 is recruited to these different membranes remains unclear. Here, we investigate how the functional regulation of the SNARE domain can mechanistically regulate Ykt6 membrane recruitment and activity in mammalian cells and in *Drosophila*.

## 2. Material and Methods

The *Drosophila* Ykt6 coding region was amplified and the PCR product was recombined into the pDONR^TM^221 vector using the Gateway BP Clonase II Enzyme mix (Life Technologies, Carlsbad, CA, USA). Point mutations of potential phosphorylation sites (S175, S182, T188, T192) were introduced by site-directed mutagenesis. For the generation of transgenic flies, constructs were subcloned into expression vectors pUASt-attB-rfA-mCherry and pUASt-attB-mCherry-rfA (kind gift from Sven Bogdan) by LR recombination (Life Technologies, Carlsbad, CA, USA). Human Ykt6 was amplified from hYkt6-Myc (C-Terminal myc-destination plasmids (DKFZ—Genomics and Proteomics Core Facility)) and the PCR product was inserted into pcDNA3.1MycBioID (Addgene #35700). Point mutations for Ykt6-3A (S174A, T181A, S187A), Ykt6-3E (S174E, T187E, S181E), F42A, C194A, C195A, and relevant combinations were introduced by site-directed mutagenesis. MycBioID tag was removed via Nhe1/Xho1 to obtain untagged constructs in pcDNA3.1. RUSH-EGFP-Wnt3A was constructed by amplifying the core protein sequence of Wnt3A and integrating it by Gibson cloning [21] with the Wnt3A signal peptide and the streptavidin-binding peptide sequence into the ER-hook containing the pCMV-KDEL-IVS-IRES-reporter plasmid backbone [22]. The following expression constructs were used: pCMV-Wnt3A [23], DsRed-Rab5-QL (E. De Robertis, Addgene #29688) (Table 1)

### 2.1. Antibodies

Antibodies were used against Calnexin, 1:1000, (WB, rabbit), Dallas, Texas, US; CD81 1:1000 (1.3.3.22, WB, mouse (DLN-09707), Dianova, Hamburg, Germany; EEA1, 1:300 (IF, mouse (610456), BD, New Jersey, NJ, USA; GAPDH (6C5), 1:5000 [WB; mouse (AM4300)], Ambion, Austin, TX, USA; Hsc70 1:2000 (WB; mouse (sc-7298), Santa Cruz, CA, CA, USA; TSG101, 1:1000, (WB, rabbit (HPA006161), Sigma, MO, USA; Wnt3A, 1:500 (WB, rabbit), Abcam, Cambridge, UK, and Ykt6 WB and IF; mouse (sc-365732), Santa Cruz, CA, USA. Antibodies against Ykt6 were generated by immunizing two guinea pigs with the peptides KVSADQWPNGTEATI (aa 105–119, within Longin domain) and YQNPVEADPLTKMQN (aa 131–145, covers part of the SNARE domain). Final bleeds were pooled and affinity purified against the original peptides (Eurogentec). Secondary antibodies directed against the species of interest were coupled to Alexa Fluor 488, 568, 594 and 647 IF, 1:500, Invitrogen, Carlsbad, CA, USA and 680RD and 800CW WB, 1:20,000, LiCor, Lincoln, NE, USA.

### 2.2. Drosophila Stocks and Genetics

The following Drosophila stocks were used in this study: *en-GAL4, UAS-GFP* (chr. II, a gift from J. Grosshans). The following stocks were obtained from the Bloomington Drosophila stock center: *UAS-Dcr; enGAL4, UAS-GFP* (#25752), *tub-GAL80TS* (#7108), and *vas-PhiC31; attP.ZH-86Fb* (#24749). The Ykt6 (KK105648) UAS-RNAi stock was obtained from the Vienna Drosophila RNAi Center. UAS-Ykt6 transgenic lines were generated according to standard protocols by φC31 integrase-mediated site-specific insertion in the attP landing site at ZH-86Fb [24]. Fly stocks were kept on standard medium containing agar, yeast, and corn flour. Crosses were performed at 25 °C or RT.

### 2.3. Kinase Screen

Biotinylated Ykt6-WT peptide (GEKLDDLVSKSEVLGTQSKAFYKTARKQN) was tested in one concentration against 245 Ser/Thr kinases in a radiometric, FlashPlate PlusTM-based assay. Ten out of 18 kinase hits identified among the 245 kinases were subsequently confirmed in a hit confirmation experiment. For this, three peptide concentrations in triplicate for each of the 10 tested kinase hits were used for N-terminally biotinylated peptides of Ykt6-WT and Ykt6-3E (GEKLDDLVSK**E**EVLGTQ**E**KAFYK**E**ARKQN). The Kinase screen was performed by Reaction Biology, available online: https://www.proqinase.com/products-service-biochemical-assay-services/kinasefinder, received on 15th October 2018).

### 2.4. Cell Culture and Transfection

Hek293T and HCT116 cells were maintained in DMEM (Gibco) supplemented with 10% fetal calf serum (Biochrom) at 37 °C in a humidified atmosphere with 5% CO_2_. Cells were transiently transfected with Screenfect siRNA for siRNA and Screenfect A (Screenfect) for plasmids according to the manufacturer’s instructions. Cells were identified and checked regularly for mycoplasma contamination.

### 2.5. Blue Sepharose Precipitation

The relative amount of Wnts secreted into cell culture supernatant was analyzed using Blue Sepharose precipitation as described [23,25]. Shortly, HEK293T cells were transiently transfected in 6-well plates with 1 µg of Wnt3A plasmids. Then, 72 h after transfection, the supernatant was collected and centrifuged at 4000× *g* rpm to remove cell debris, transferred to a fresh tube, and rotated at 4 °C for 1 h with 1% Triton X-100 and 40 µL of Blue Sepharose beads. The samples were washed and eluted from the beads using 2X SDS buffer with β-mercaptoethanol and analyzed by immunoblotting.

### 2.6. Extracellular Vesicle purification

Extracellular vesicles were purified by differential centrifugation as described previously [26,27]. In short, supernatants from mammalian cells were subjected to sequential centrifugation steps of 750× *g*, 1.5 × 10^3^ g and 1.4 × 10^4^ g, before pelleting exosomes at 1× 10^5^ g in a SW41Ti swinging bucket rotor for 2 h (Beckman). The supernatant was discarded, and exosomes were taken up in 1/100 of their original volume in H_2_O.

### 2.7. Immunostainings, Microscopy, and Image Analysis

For immunofluorescence staining, cells were reverse transfected with siRNAs, seeded in 6 well dishes or 8-well microscopic coverslips, 24 h later transfected with indicated plasmids, and 48–72 h later fixed with 4% paraformaldehyde. Cells were permeabilized with 0.1% Triton X-100 and blocked in 10% BSA/PBS. Primary antibodies in PBS were incubated for 1 h at room temperature and antibody binding was visualized by fluorochrome-conjugated secondary antibodies. Confocal images were processed with Zen lite (Zeiss, Oberkochen, Germany), Fiji/ImageJ (NIH, Rockville, Maryland, MA, USA) [28,29,30] and Affinity Designer (Affinity Serif, San Francisco, CA, USA).

### 2.8. Rab5QL Assay and Quantification

Hek293T cells were co-transfected with plasmids for Rab5Q79L-DsRed and either control, Ykt6-WT or Ykt6-3E and analyzed by immunofluorescence microscopy, and the size of enlarged Rab5Q79L-positive endosomes was measured in different biological replicates.

### 2.9. Membrane Fractionation

As previously described [31], HEK293T cells were seeded and transfected with Ykt6-WT plasmid. Then, 48 h post transfection, cells were lysed on ice with 1 mL of Lysis buffer A (150 mM NaCl, 50 mM HEPES, 0.1% Saponin, 1 M Glycerol, and 1% PIC) and then centrifuged at 2000× *g* for 10 min at 4 °C; then, the supernatant (cytosolic fraction) was transferred to a new tube. The pellet was lysed in 1 mL of Lysis Buffer B (150 mM NaCl, 50 mM Hepes, 1% Igepal, 1 M Glycerol and 1% PIC) and incubated rotating for 30 min at 4 °C. Then, after being centrifuged at 7000× *g* for 10 min at 4 °C, the supernatant was transferred to a new tube (membrane fraction).

### 2.10. BioID Pull Down and Mass Spectrometry 

For large-scale BioID pull down, cells were seeded and 24 h later transfected with BioID-WT or mock constructs. Then, 36 h post transfection, 50 μM biotin was added over night. Cells were washed with PBS twice, cell fractionated, and then boiled 5 min in non-reducing SDS sample buffer (300 mM Tris-HCl pH 6.8, 12% SDS, 0.05% Bromphenolblue, 60% Glycerol, 12 mM EDTA), run a short-distance (1.5 cm) on a 4–12% NuPAGE Novex Bis-Tris Minigel (Invitrogen). Gels were stained with Coomassie Blue for visualization purposes. Full lanes were sliced into 23 equidistant slices regardless of staining, short runs cut out as a whole and diced. After washing, gel slices were reduced with dithiothreitol (DTT), alkylated with 2-iodoacetamide, and digested with trypsin overnight. Then, the resulting peptide mixtures were extracted, dried in a SpeedVac, reconstituted in 2% acetonitrile/0.1% formic acid/(*v*:*v*), and prepared for nanoLC-MS/MS as described previously [32].

For the generation of a peptide library for SWATH-MS, equal amount aliquots from each sample were pooled to a total amount of 80 μg and separated into eight fractions using a reversed phase spin column (Pierce High pH Reversed-Phase Peptide Fractionation Kit, Thermo Fisher Scientific, Waltham, Massachusetts, United States. MS analysis Protein digests were separated by nanoflow chromatography. Then, 25% of gel slices or 1 μg aliquots of digested protein were enriched on a self-packed precolumn (0.15 mm ID × 20 mm, Reprosil-Pur120 C18-AQ 5 μm, Dr. Maisch, Ammerbuch-Entringen, Germany) and separated on an analytical RP-C18 column (0.075 mm ID × 250 mm, Reprosil-Pur 120 C18-AQ, 3 μm, Dr. Maisch) using a 30 to 90 min linear gradient of 5–35% acetonitrile/0.1% formic acid (*v*:*v*) at 300 nl/ min.

SWATH-MS library generation was performed on a hybrid triple quadrupole-TOF mass spectrometer (TripleTOF 5600+) equipped with a Nanospray III ion source (Ionspray Voltage 2400 V, Interface Heater Temperature 150 °C, Sheath Gas Setting 12) and controlled by Analyst TF 1.7.1 software (SCIEX, Framingham, Massachusetts, MA, USA)build 1163 (all AB Sciex), using a Top30 data-dependent acquisition method with an MS survey scan of *m*/*z* 380–1250 accumulated for 250 ms at a resolution of 3.5 × 10^4^ full width at half maximum (FWHM). MS/MS scans of *m*/*z* 180–1500 were accumulated for 100 ms at a resolution of 17,500 FWHM and a precursor isolation width of 0.7 FWHM, resulting in a total cycle time of 3.4 s. Precursors above a threshold MS intensity of 200 cps with charge states 2+, 3+, and 4+ were selected for MS/MS, and the dynamic exclusion time was set to 15 s. MS/MS activation was achieved by CID using nitrogen as a collision gas and the manufacturer’s default rolling collision energy settings. Two technical replicates per reversed phase fraction were analyzed to construct a spectral library.

For quantitative SWATH analysis, MS/MS data were acquired using 100 variable size windows [33] across the 400–1200 *m*/*z* range. Fragments were produced using rolling collision energy settings for charge state 2+, and fragments acquired over an *m*/*z* range of 180–1500 for 40 ms per segment. Including a 250 ms survey scan, this resulted in an overall cycle time of 4.3 s. Two replicate injections were acquired for each biological sample.

### 2.11. Mass Spectrometry Data Processing

For SWATH-MS analysis, protein identification was achieved using ProteinPilot Software version 5.0 (SCIEX, Framingham, Massachusetts, MA, USA) build 4769 (AB Sciex) at “thorough” settings. MS/MS spectra from the combined qualitative analyses were searched against the UniProtKB Homo sapiens reference proteome (revision February 2017. 92,928 entries) augmented with a set of 51 known common laboratory contaminants to identify 597 proteins at a False Discovery Rate (FDR) of 1%. Spectral library generation and SWATH peak extraction were achieved in PeakView Software version 2.1 (SCIEX, Framingham, Massachusetts, MA, USA) build 11041 (AB Sciex) using the SWATH quantitation microApp version 2.0 SCIEX, Framingham, Massachusetts, MA, USA) build 2003. Following retention time correction on endogenous peptides spanning the entire retention time range, peak areas were extracted using information from the MS/MS library at an FDR of 1% [34]. The 26 resulting peak areas were summed to peptide and protein area values, which were used for further statistical analysis. Reactome Functional Network analysis [35] was performed with Cytoscape [www.cytoscape.org (Accessed on 15th June 2020)] and Kegg pathway analysis was performed with David [36].

### 2.12. Statistics

All experiments were carried out at least in biological triplicates. Error bars indicate s.d. Statistical significance was calculated by carrying out Student’s t-test where appropriate or one-way ANOVA with Dunnett’s multiple comparison test to compare a control mean with the other means.

## 3. Results

### 3.1. Several Phosphorylation Sites in the Ykt6 SNARE Domain Are Evolutionarily Conserved

Specific phosphorylation sites within the SNARE domain of non-neuronal SNAREs are conserved over the plant, fungi, and animal kingdoms [37]. These sites are located within the SNARE layers and sterically block the interacting domains of the helices. Their importance was demonstrated using the example of the SNARE VAMP8, for which the phosphorylation or mutation of these sites inhibits the fusion of secretory granules [37]. To identify these conserved residues in Ykt6, we aligned its protein sequences deduced from the genomes of *S. cerevisiae*, *C. elegans*, *D. melanogaster,* and vertebrates such as *D. rerio*, *M. musculus,* and *H. sapiens* (Figure 1A). Within the largely hydrophobic SNARE alpha-helix, residues at helical layer positions are designated from 0 to +8 starting from the ionic residue (Arginine or Glutamine) toward the C-Terminus [3]. We found two conserved Serines and a Threonine (S174, S181, and T187 in the human sequence) at layers +3 and +5 and +7, respectively. Furthermore, an additional Threonine (T192) was identified in *Drosophila* at layer +8 (Figure 1A). In confirmation, position S174 was identified in different published phosphoproteomic approaches [38]. Prediction according to NetPhos 3.1 [39] suggested that the identified phosphorylation sites are potential CDK1 and PKC sites (Figure 1B). In light of the identified conserved phosphorylation sites, we decided to study the mechanistic role of possible phosphorylations in human and *Drosophila* Ykt6 in more detail (Figure 1C). To identify potential kinases that phosphorylate these sites, a biotinylated peptide of the human Ykt6 SNARE domain was subjected to a bead-based kinase screen with 245 Serine/Threonine-kinases (Figure 1D). Out of 18 kinases that phosphorylated the peptide more than two-fold over background (Appendix A), ten were selected (Figure 1E) and subjected to a validation screen. In comparison to the human Ykt6-WT peptide, a Ykt6 peptide with three sites mutated to Glutamic acid (S174E, S181E, T187E, termed Ykt6-3E) was used to test for the specificity of these three positions. In vitro, phosphoinositide-dependent kinase-1 (PDK1) phosphorylated the Ykt6-WT SNARE domain and did not phosphorylate the Ykt6-3E peptide over the background signal (Figure 1F).

To test the physiological relevance of the identified Ykt6 phosphorylation sites, we expressed Ykt6 RNAi and overexpressed Ykt6 rescue constructs in *Drosophila* with the UAS/GAL4 system [40]. While a strong Ykt6 knockdown by enGAL4-driven UAS-Ykt6 RNAi at 25 °C was lethal at the larval stage, this phenotype was rescued by the overexpression of Ykt6-WT and a non-phosphorylatable Ykt6-4A (S175A, S1812A, T188A, T192A) mutant, but not by the overexpression of a phosphomimicking Ykt6-4E (S175E, S1812E, T188E, T192E) (Table 2, right column). Similarly, a mild enGAL4-driven Ykt6 knockdown at RT showed (1) melanotic tumors in larvae, a sign of increased cell death in combination with phagocytic clearance activity [41], and (2) wing notches in adult flies, typical cell growth, and Wnt signaling defects [42,43] In both cases, Ykt6-WT and -4A expression rescued these phenotypes, but Ykt6-4E did not (Figure 1G, middle column). Taken together, these results indicate that Ykt6 has evolutionarily conserved phosphorylation sites within the SNARE domain, which are important for Ykt6 function in cell growth.

### 3.2. Phosphomimicking Mutations Accumulate Ykt6 at Membranes in the Secretory Pathway

To investigate the mechanistic relevance of phosphorylation of the Ykt6 SNARE domain, we checked the cellular localization of Ykt6–WT or -3E by immunofluorescence microscopy. Transiently expressed constructs in HCT116 cells showed a diffuse cytoplasmic localization for Ykt6-WT, while Ykt6-3E distinctly localized to the perinuclear area and the plasma membrane (Figure 2A), suggesting a membrane association of Ykt6-3E. Co-staining with the Golgi marker GM130 revealed that Ykt6-3E strongly colocalizes with the Golgi as seen in intensity line profiles (Figure 2B), which is in line with the function of Ykt6 in the organization of Golgi apparatus [7].

Ykt6 was shown previously to be involved in Wnt secretion from *Drosophila* cells of the developing wing epithelium as well as from human cells [23,44]. To understand the dynamics of Ykt6-membrane attachment, we used Wnt3A secretion as a Ykt6-dependent process to study the role of the putative phosphorylation sites. We used “retention using selective hook” (RUSH), which is an inducible system for the release of secretory cargo [22]. This system consists of an ER-resident streptavidin-KDEL fusion protein (“hook”) and a “bait”-protein fused to a streptavidin-binding peptide (SBP) to be retained in the ER in the absence of biotin (Figure 2C). We constructed a tagged Wnt3A with an integrated GFP and an SBP between the signal peptide and the core sequence of Wnt3A. In Hek293T cells, this construct was secreted into the supernatant in the presence of biotin (Figure 2D) and localizes to the ER in the absence of biotin (Figure 2E). An addition of 50 µM biotin triggered the ER-release of Wnt3A by competitive binding to the streptavidin hook, and 30 to 60 min later, Wnt3A localized to the perinuclear region, indicating transport to the Golgi (Figure 2E,F). In cells co-expressing Ykt6-WT, Wnt3A is detected both at the Golgi and in the cytoplasm at later time points (>120 min), partially co-localizing with Ykt6 (Figure 2E,G). In contrast to Ykt6-WT, the co-expression of Ykt6-3E leads to strong accumulation with Wnt3A at the plasma membrane (>120 min) (Figure 2F,H). Hence, in the presence of endogenous Ykt6, the phosphomimicking Ykt6 mutant neither has a dominant negative effect, nor blocks ER to Golgi trafficking of Wnt3A but accumulates at different post-Golgi membranes within the secretory pathway together with Wnt3A.

### 3.3. SNARE Phosphorylation Sites Determine Membrane Attachment and Autoinhibited Conformation

To further analyze the role of Ykt6 phosphorylation in membrane attachment, we used a differential detergent fractionation [31] to biochemically separate membranes from cytoplasm.

Confirming immunofluorescence results (Figure 2A), we found that Ykt6-3E partially associated with the membrane fraction, whereas Ykt6-WT was exclusively found cytoplasmically (Figure 3A, left panel, lane 5 and 7). In addition, different Ykt6 mutants previously described to impair Ykt6 conformational changes were also found only in the cytoplasmic fraction. This includes mutations of either palmitoylation (C194A) or farnesylation (C195A) sites, the palmitoyl/farnesyl (C194/195A) double mutant, and a combination of phosphomimicking and palmitoyl/farnesyl (Ykt6-3E/C194A/C195A) mutants (Figure 3A). This affirms that farnesylation and subsequent palmitoylation is required for the stable membrane association of Ykt6 [5,17]. Mutation in the Longin domain (F42A) was shown to impair the auto-inhibited conformation by shielding the lipidation [6]. Indeed, we detected some Ykt6-F42A in the membrane fraction, while non-phosphorylatable Ykt6-3A did not show this membrane association (Figure 3A, right panel, lane 11 and 13). All constructs were expressed at levels comparable to endogenous Ykt6 in total cell lysate, excluding any effect of strong overexpression (Figure 3B). This demonstrates that phosphomimicking modifications within the SNARE domain result in a form of Ykt6 that preferentially associates with membranes, possibly through inhibition of its closed conformation.

To test whether Ykt6 expression levels might determine the level of membrane recruitment, we expressed increasing amounts of WT constructs (0.1, 0.3, and 1 µg). This leads to an expected linear increase of Ykt6 expression (Figure 3C) in the overall cell lysates. While this linear increase is also visible in the cytoplasmic fraction (Figure 3D), the level of Ykt6-3E, but not Ykt6-WT is strongly increased in the membrane fraction, (Figure 3E). This indicates that it is not the cellular level of Ykt6 determines membrane recruitment but specifically the modification of its SNARE domain. 

Next, we used an unbiased BioID approach [45] to label proteins in close vicinity of Ykt6 by fusion with the prokaryotic BirA* domain (Figure 3F). N-terminally tagged Ykt6-WT and mutant constructs were expressed at comparable levels (Figure 3G) in the presence of 50 µM biotin. Biotinylated proteins were purified by streptavidin pulldown and subjected to mass spectrometry to define the “proxisome” of the functional Ykt6 SNARE domain (Figure 3F). We found Ykt6 highly enriched among the biotinylated proteins compared to background proteins labeled by the BioID alone, indicating an ability of the N-terminal BioID domain to intra- or intermolecularly label Ykt6. We hypothesized that the closed conformation of Ykt6 could interfere with self-labeling and compared the biotin-labeling ability of Ykt6-3E with different Ykt6 point mutations known to interfere with the closed conformation (Figure 3H). Both cytosolic and membrane fractions were analyzed by immuno-blotting for streptavidin-labeled Ykt6-BioID. Compared to Ykt6-WT, the phosphomimicking (3E) and the palmitoyl/farnesyl (C194/195A) double mutant BioID constructs were strongly labeled by the N-terminal BioID domain in the cytoplasmic fraction (Figure 3H, left panel, lane 6, and 12 and quantification in (I). Minor self-labeling of Ykt6-WT, due to binding of the functional Longin domain to the farnesyl group, suggests that a large portion acquires a closed conformation, which is in agreement with the lower Ykt6-WT levels in the membrane fraction (Figure 3A, lane 5). In contrast, combined mutations of palmitoylation/farnesylation sites and the phosphomimicking mutations (Ykt6-3E/C194A/C195A) lead to a more unstable and open conformation, allowing stronger self-labeling by the N-terminal BioID (Figure 3H,I). Taken together, these complementary approaches suggest that modification of the Ykt6 SNARE domain can prevent Ykt6 from assuming a closed auto-inhibited conformation and thereby promote Ykt6 membrane recruitment.

### 3.4. Ykt6 SNARE Domain-Dependent Proximity Proteome

Ykt6 is involved in the homotypic fusion of ER and vacuolar membranes, retrograde Golgi trafficking in yeast, and autophagosome formation [12,15,19,46]. However, the mechanism of how Ykt6 is recruited to those different organelles is not well understood. As the phosphomimicking Ykt6-3E construct was present in the membrane fraction and was labeled strongly by the N-terminal BioID domain compared to Ykt6-WT, we next subjected the pull-down of membrane and cytoplasmic fractions to mass spectrometry in order to identify biotinylated proteins in both fractions. We reasoned that although Ykt6-WT steady-state membrane levels were low, BioID labeling might label transient Ykt6-WT interaction partners at membranes during its cycling. Ykt6-WT and -3E were detected at similar levels among biotinylated proteins of the cytoplasmic fraction, but Ykt6-3E was approximately six-fold enriched over Ykt6-WT in the membrane fraction (Appendix A). We identified a total of 241 enriched proteins in Ykt6-WT and 235 enriched proteins in -3E, indicating that labeling by both constructs was similarly successful. Then, we performed an enrichment analysis based on PANTHER classification of the identified proteins [47]. At first, we compared the enriched biological processes of Ykt6-WT between the cytoplasmic and the membrane fraction (Figure 4A–C). Among the biological processes exclusive to the membrane fractionated Ykt6-WT proxisome were vesicle trafficking, regulated exocytosis, organelle organization, and various biosynthetic processes (Figure 4C). This further infers that the membrane-bound Ykt6 acts as an active SNARE proximal to very different cellular processes.

Next, we compared the processes differentially identified in the proxisomes of theYkt6-WT and -3E of membrane fractions (Figure 4D–H). GO term enrichment and Reactome pathway analysis showed Ykt6-WT to be proximal to processes such as Golgi-ER traffic and autophagy, among others. In contrast, Ykt6-3E was proximal to Akt and Wnt Signaling, protein folding, and protein ubiquitination (Figure 4F,H). This suggests that modification of the Ykt6 SNARE domain plays a role in the functional membrane attachment of Ykt6 in specific cellular pathways.

### 3.5. Ykt6 Regulates EV Secretion in a Concentration-Dependent Manner

Due to the role of Ykt6 in autophagosome formation [15,18,19] and extracellular vesicle (EV) secretion [23] we next wanted to address how Ykt6 conformational changes via the phosphorylation sites in the SNARE domain affect endosomal sorting and EV secretion. Exosomes are a population of small extracellular vesicles. They are generated as intraluminal vesicles (ILVs) by inward budding of the limiting membrane of multivesicular bodies (MVBs) and can be secreted in an ESCRT- as well as an Alix-Syntenin-dependent manner [48]. Microautophagy, which is involved in the degradation of soluble cytosol, occurs during MVB formation and shares the ESCRT components for vesicle formation and cargo internalization [49]. To understand whether Ykt6 membrane recruitment plays a role at the level of cargo sorting into ILVs, we used constitutively active Rab5 (Rab5Q79L) to enlarge endosomes and thereby visualize whether cargo is present at the limiting membrane or within MVBs [50]. In Rab5Q79L-expressing Hek293T cells, overexpressed Ykt6-WT as well as -3E localized inside enlarged endosomes (Figure 5A). However, in Ykt6-3E overexpressing cells, those endosomes were significantly smaller than in Ykt6-WT (Figure 5B), indicating that the mutated SNARE domain might hinder endosomal fusion events activated by Rab5Q79L. Interestingly, under serum-fed conditions, Ykt6 had no effect on the levels of p62, as a readout of autophagy induction (Figure 5C), suggesting that the non-canonical role of Ykt6 in autophagy could be starvation-dependent [51,52].

A portion of ILVs is secreted as exosomes by the fusion of MVBs with the plasma membrane. Thus, we next compared the effect of Ykt6 on EV secretion into the supernatant of Hek293T cells. Knockdown of Ykt6 as well as Alix reduced the level of EVs as measured by nanoparticle tracking analysis (NTA) after differential ultracentrifugation for EV isolation (Figure 5D,F,G). Surprisingly, similar to Ykt6 knockdown, the overexpression of tagged Ykt6-WT significantly reduced secretion of EVs, but Ykt6-3E did not (Figure 5E,H,I). This suggests that the amount of Ykt6 regulates its function in secretory processes and further highlights that in line with the effect observed on Rab5Q79L endosomes, Ykt6 function can be impaired by modification of the Ykt6 SNARE domain. Ykt6 knockdown was previously shown to reduce Wnt as well as exosome secretion [23]; thus, we next checked Wnt levels on exosomes (P100) purified from the supernatant of Hek293T cells. Here, the overexpression of Ykt6-WT as well as Ykt6-3E reduced the level of Wnt3A (Figure 5J). Membrane recruitment of Ykt6 was not increased with higher levels of Ykt6-WT (Figure 3E). However, looking at the secretion of Wnt3A, we found that increasing amounts of Ykt6-WT decreased the level of Wnt3A on exosomes to a level similar to Ykt6 knockdown (Figure 5K). This could imply that the phosphorylation and fusion competence of Ykt6 present at endosomal membranes is a concentration-dependent process in exosome secretion.

## 4. Discussion

The fusion of eukaryotic transport vesicles with target organelles requires membrane-bridging complexes of membrane anchored SNAREs. Ykt6 is lacking a transmembrane domain, and thus, its site of action is determined by changing from a soluble to a membrane-bound conformation. How Ykt6 is recruited to membranes remains unclear. Here, we investigate how the functional regulation of its SNARE domain can mechanistically regulate Ykt6 membrane attachment and its activity in mammalian cells and in *Drosophila*. We found that Ykt6 membrane attachment is regulated by modifications in its SNARE domain, and that this regulation affects cell survival in vivo. Ykt6 phosphorylation and attachment further affect the Ykt6 proximity profile and the cellular processes in which it takes part. Specific secretory processes, such as Wnt and EV secretion, furthermore depend on a functional Ykt6 SNARE domain and could thus be regulated by Ykt6 phosphorylation. Under normal growth conditions, these processes seem to be more important for cellular function, than its role in autophagosome formation. As it has also been reported that Ykt6 is involved in autophagosome–lysosome fusion in human cells and *Drosophila* fat body under starvation-induced conditions [15,19], it is possible that Ykt6 is allocated to alternate compartments, especially under nutrient stress condition [51,52].

Non-neuronal SNAREs possess evolutionarily conserved phosphorylation sites [37], which seem to prevent membrane fusion when SNAREs are phosphorylated. Based on our data, Ykt6 functions in a similar manner, as Ykt6-3E seems to prevent fusion as well. Yet, dynamic membrane recruitment is specific to SNAREs lacking a transmembrane domain such as Ykt6. Indeed, self-labeling and membrane attachment of different Ykt6 mutants varies and in agreement with previous work [53] hints toward a two-step process of activation that is specific to Ykt6 and involves (1) a change from closed to open confirmation and (2) membrane recruitment. Post-membrane recruitment, true for all SNAREs with phosphorylation sites in the SNARE domain [37], phosphorylation would prevent the fusion of different SNAREs to mediate membrane fusion.

Protein acylation is a regulatory post-translational modification regulating membrane association and the dissociation of its target proteins. The F42 position has been previously shown to participate in intramolecular interactions between the Ykt6 Longin domain and the SNARE motif. The hydrophobic face at position F42 within the Longin domain seems to accommodate the farnesyl anchor, as seen in a crystal structure with dodecylphosphocholine (DPC) [7]. This intramolecular binding explains the membrane association of Ykt6 when F42 is mutated (Figure 3A, right panel, lane13). Similarly, the phosphorylation sites we identified (S174/S181/T187) are directly facing this interface and therefore likely impact the tight folding of Ykt6 into a closed conformation and thus membrane attachment (Figure 3E). Interestingly, the non-phosphorylatable mutant Ykt6-3A does not attach to membranes by itself, yet it is able to rescue the growth defects in vivo (Figure 1G). Taken together, this suggests that the phosphorylation of Ykt6 prevents the conformational switch to the inactive state, which leads to membrane stabilization. Likely, an additional step is required to render Ykt6 fusion-competent, as Ykt6-3E did not rescue Ykt6 KD in vivo*,* but it had no dominant-negative effects in the presence of endogenous Ykt6. This last step toward fusion-competent Ykt6 could be achieved by dephosphorylation [52], for example by calcineurin, as suggested by a recent study [54].

Which signaling pathways could stimulate the phosphorylation of Ykt6 and therefore its membrane recruitment? As PKC gets activated in the context of endocytosis to recruit adaptor complexes to endosomes [55,56] local PKC-dependent conformation changes and subsequent palmitoylation could stimulate Ykt6 association with endosomes. Similarly, PDK1 is a growth factor-dependent kinase that could phosphorylate Ykt6 in the presence of mitogenic signals that require Wnt secretion and/or autophagy. Moreover, the Netphos prediction of Ykt6 S174 to be a CDK1 target and the fact that proteins proximal of Ykt6 are involved in cell cycle regulation opens interesting questions of whether Ykt6 is involved in cell growth-dependent regulations of organelle fusion. Furthermore, the identification of three phosphorylation sites bears the possibility that the three sites are individually targeted by different signaling pathways and differ in their role in recruiting and stabilizing Ykt6 to and at membranes under permissive circumstances. Yet, the physiological relevance of Ykt6 phosphorylation remains to be demonstrated, and more detailed analysis is necessary to understand the interconnection and hierarchy of different signaling pathways targeting Ykt6.

Interestingly, other SNAREs such as Syntaxin-5 are modified by monoubiquitination that regulates Golgi integrity during cell cycle progression [57]. In addition to position S174 in Ykt6 being identified in Phosphoproteomic studies, its Lysine 182 and 186 (K182,K186) were found to be ubiquitinated in different human cell lines and the mouse liver [58,59,60]. These positions lie close by the analyzed phosphorylation sites (S181, T187). Thus, Ykt6 function and its membrane recruitment could be the target of different posttranscriptional modifications at the crossroad of cellular trafficking events in cellular growth.

## Figures and Tables

**Figure 1 biomolecules-10-01560-f001:**
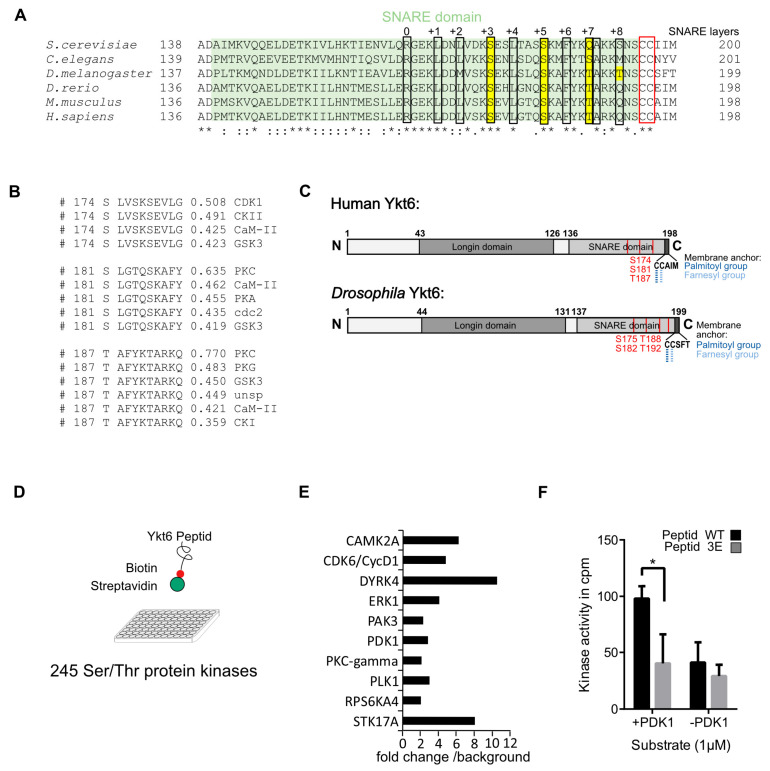
Several phosphorylation sites in the Ykt6 sensitive factor attachment protein receptors (SNARE) domain are evolutionarily conserved. (**A**) Sequence alignment of Ykt6 SNARE domain (Human: position 136–198) of different eukaryotes. Conserved serines and threonines at SNARE layers are marked in yellow, lipid-modified cysteines are in the red box. (**B**) PhosphoSite prediction for human Ykt6 position S174, S181, and T187 using Netphos 3.1 [39]. (**C**) Scheme of human and *Drosophila* Ykt6 with predicted phosphorylation sites that were either mutated to alanine or glutamic acid. (**D**) In vitro kinase assay of biotinylated peptide of the Ykt6-WT SNARE domain with 245 different serine/threonine kinases. (**E**) Ten out of 18 identified kinases phosphorylating Ykt6 SNARE domain >twofold over background were validated. (**F**) A validation kinase assay comparing Ykt6-WT versus Ykt6-3E found phosphoinositide-dependent kinase-1 (PDK1) as a kinase phosphorylating the SNARE domain. * = 0.05.

**Figure 2 biomolecules-10-01560-f002:**
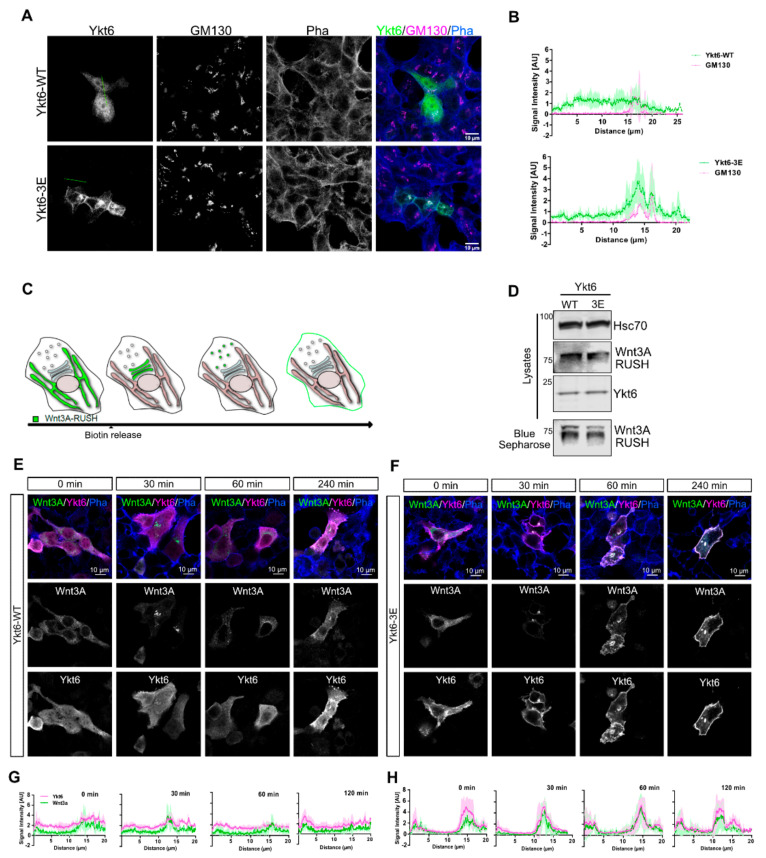
Phosphomimicking mutations accumulate Ykt6 at membranes in the secretory pathway. (**A**) Representative immunofluorescence and (**B**) Quantification of Ykt6-WT and -3E transfected in HCT116 cells and additionally stained for GM130, Phalloidin, and Hoechst. Scale bar 10 µm. (**C**) Scheme representation of Wnt3A-RUSH (“retention using selective hook”) system. (**D**) Representative immunoblot of Wnt3A, Ykt6 WT, and -3E in Hek293T lysates and blue sepharose precipitation of Wnt3A secretion from the supernatant after 24 h of biotin treatment. (**E**) HCT116 cells transfected with RUSH-EGFP-Wnt3A in combination with Ykt6-WT and quantification in (**G**). (**F**) Ykt6-3E stained for Ykt6 and F-actin in addition to GFP and quantification in (**H**). Scale bar represents 10 µm, representative images, and quantifications of three independent experiments.

**Figure 3 biomolecules-10-01560-f003:**
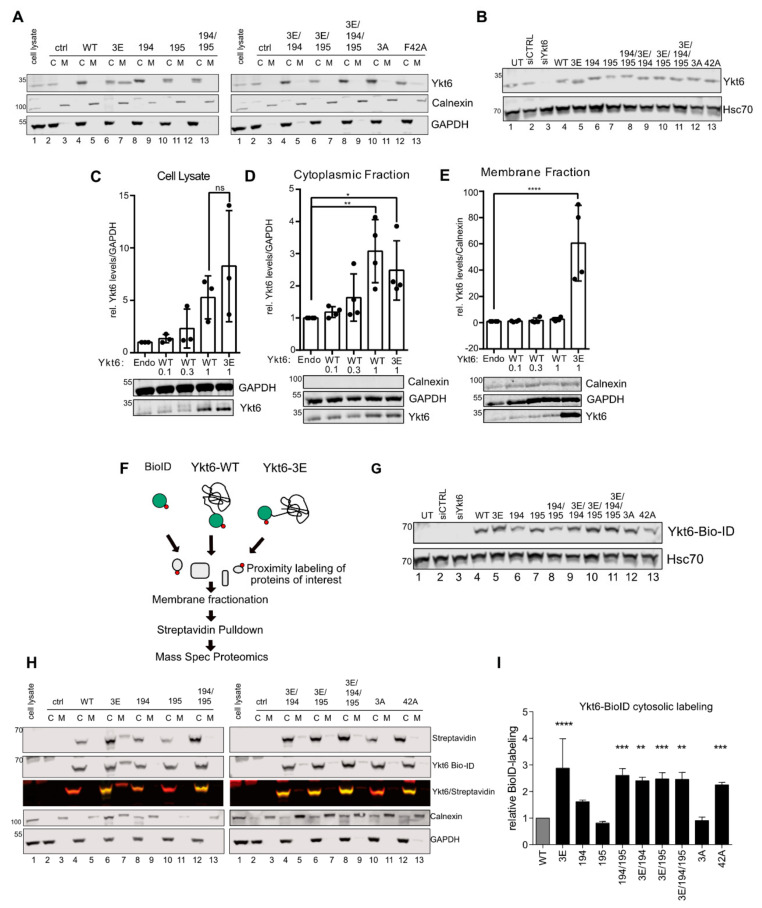
SNARE phosphorylation sites determine membrane attachment and auto-inhibited conformation. (**A**) Representative immunoblot of cell fractionation of different Ykt6 mutant constructs transfected in Hek293T cells and stained with Ykt6 and fraction markers GADPH for cytoplasmic (C) and Calnexin for membrane (M) fraction (*n* = 4). (**B**) Representative immunoblot of Ykt6 knockdown and overexpression of different Ykt6 mutant constructs with Hsc70 as a loading control. (**C**) Representative immunoblot and quantification of overexpressed Ykt6-WT construct in three different concentrations (0.1 µg, 0.3 µg, and 1 µg), along with overexpressed Ykt6-3E construct and endogenous Ykt6 from three biological replicates. (**D**) Representative blot and quantification of endogenous and overexpressed Ykt6-WT (0.1 µg, 0.3 µg, and 1 µg), 3E (1 µg) in the cytoplasmic fraction, n = 4 (**E**) Representative blot and quantification of endogenous and overexpressed Ykt6-WT (0.1 µg, 0.3 µg, and 1 µg) or 3E (1 µg) in the membrane fraction, *n* = 4. (**F**) Scheme of biotin-labeling assay. Biotin–ligase BioID was expressed alone or as Ykt6-WT or -3E fusion constructs in Hek293T cells in the presence of 50 µM biotin. After membrane fractionation, proteins were streptavidin-purified and subjected to mass spectrometry or immunoblotting. (**G**) Representative immunoblot of Ykt6 knockdown and overexpression of different Bio-ID-tagged Ykt6 mutant constructs with Hsc70 as a loading control (**H**) Representative immunoblot of cell fractionation of different Ykt6-BioID constructs transfected in Hek293 cells, stained with Streptavidin, Ykt6 and fraction markers as in (**B**) (*n* = 3). (**I**) Quantification of biotin labeling in the cytosolic fraction of (**H**). * = 0.05, ** = 0.01, *** = 0.001, **** = 0.0001.

**Figure 4 biomolecules-10-01560-f004:**
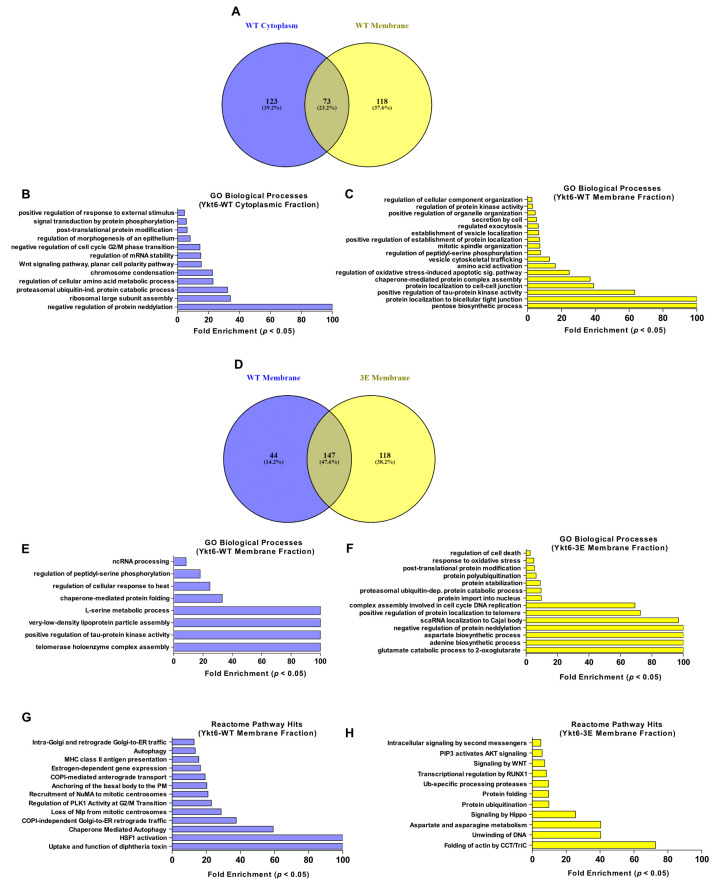
Ykt6 SNARE-dependent proximity proteome. (**A**) Proteins enriched in Ykt6-WT cytoplasmic fraction versus membrane fraction. (**B**,**C**) Enriched GO biological processes of Ykt6-WT in cytoplasmic versus membrane fraction. (**D**) Proteins up-regulated between Ykt6-WT versus Ykt6-3E in the membrane fraction. (**E**,**F**) Enriched GO biological processes of Ykt6-WT and -3E with a (*p*-value < 0.05) as determined by Fisher Exact Test with the Benjamini–Hochberg False Discovery Rate < 0.05. (**G**,**H**) Enriched Reactome Pathway hits of Ykt6-WT and -3E in the membrane fraction based on FDR < 0.05.

**Figure 5 biomolecules-10-01560-f005:**
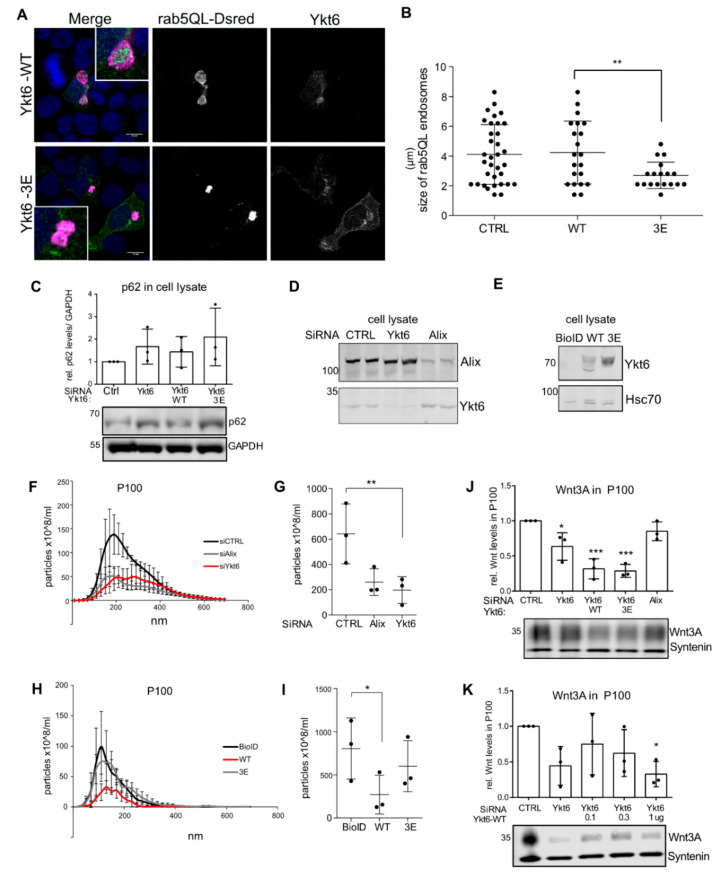
Ykt6 regulates extracellular vesicle (EV) secretion in a concentration-dependent manner. (**A**) Hek293T cells were co-transfected with plasmids for Rab5Q79L-DsRed and either control, Ykt6-WT, or Ykt6-3E and analyzed by immunofluorescence microscopy. Scale bar represents 10 µm. (**B**) Quantification of (**A**), number of endosomes: control (*n* = 34), Ykt6-WT (*n* = 21), or Ykt6-3E (*n* = 18) from two independent biological replicates. (**C**) Representative blot and quantifications of p62 levels in HCT116 cells transfected with siRNA against control or Ykt6 and Ykt6-WT or 3E. (**D**) Immunoblot of Hek293T cells transfected with siRNA against control, Ykt6, or Alix. (**E**) Immunoblot of Hek293T cells transfected with Ykt6-WT, -3E, and BioID mock plasmid. (**F**) Extracellular vesicles (EV) were purified from the supernatant. EVs in P100 are centrifuged 1 h at 100,000× *g*. Size profile of P100-EV was analyzed by nanoparticle tracking analysis (NTA). (**G**) Quantification of 100–200 nm EV size profile in (**D**) from three biological replicates. Significance level ** = 0.01. (**H**) NTA EV size profile of P100 from (**F**,**I**) Quantification of 100–200 nm sized EVs from (G) from three biological replicates. * = 0.05. (**J**,**K**) Immunoblot and quantification of Wnt3A in P100 purified from Hek293T cells transfected with siRNA against control, Ykt6 or Alix and (J) Ykt6-WT or 3E, *** = 0.001 and (**K**) Ykt6-WT in three different concentrations (0.1, 0.3, and 1 µg DNA) * = 0.05.

**Table 1 biomolecules-10-01560-t001:** Dharmacon siRNA SMARTpools.

Gene Symbol	GENE ID	Gene Accession	GI Number	Sequence
siGENOME Non-targeting Control_5 #				UGGUUUACAUGUCGACUAA
Ykt6	10652	NM_006555	34304384	GCUCAAAGCCGCAUACGAU
				GUGAGAAGCUAGAUGACUU
				GAAGGUACUAGAUGAAUUC
Alix	10015	NM_013374	371875333	GAAGGAUGCUUUCGAUAAA
				GAACAGAACCUGGAUAAUG
				GAGAGGGUCUGGAGAAUGA
				GCAGUGAGGUUGUAAAUGU

# represents the non-Targeting siRNA number provided by the company.

**Table 2 biomolecules-10-01560-t002:** Depletion of Ykt6 by RNAi (engrailed-Gal4, UAS-GFP/ UAS-ykt6 RNAi; larvae reared for three days at RT or 25 °C).

EngrailedGAL4, UAS-GFP>	Mild Knockdown (RT)		Strong Knockdown (25 °C)	
	Larvae	Adult	Larvae	Adult
ykt6 RNAi	Melanotic tumors	wing defects	lethal *	lethal
ykt6 RNAi; UAS-Ykt6-WT	Normal	viable	normal	viable
ykt6 RNAi; UAS Ykt6-4A	Normal	viable	normal	viable
ykt6 RNAi; UAS-Ykt6-4E	Melanotic tumors	wing defects	lethal	lethal

* Few larvae develop melanotic tumors.

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
