# Peer review of "Phosphorylation of Ykt6 SNARE Domain Regulates Its Membrane Recruitment and Activity"

_biomolecules, 2020, doi:10.3390/biom10111560_

Round 1
Reviewer 1 Report
the authors have now addressed many of the previous concerns raised. there are a few suggestions to the authors to correct prior to publication.
1) Figure 3 A/ B panels might be switched.
2) Figure 3 H should be reloaded. the membrane fractionation procedure seems to work in general, but this figure shows highly variable levels of calnexin. there are some lanes have barely any detectable calnexin including WT and 195. 3E has clear membrane contamination in cytosolic lane. a cleaner blot would make the results more believable to readers once published.
3) There may be a typo in line 332. please revised the sentence.
Author Response
We thank the reviewer for her/his positive evaluation. Concerning the three remaining points:
1) Figure 3 A/ B panels might be switched.
We have now swapped the panels and adjusted the text.
2) Figure 3 H should be reloaded. the membrane fractionation procedure seems to work in general, but this figure shows highly variable levels of calnexin. there are some lanes have barely any detectable calnexin including WT and 195. 3E has clear membrane contamination in cytosolic lane. a cleaner blot would make the results more believable to readers once published.
We agree that the levels of calnexin in the blots are variable, but as we do not see a carry-over of Ykt6 from cytoplasmic into membrane fraction for the constructs, we believe in the validity of our results. In the short revision time, a repetition of the experiment was not feasible. Since the main focus of the study is the membrane recruitment of Ykt6-3E, we repeated (major revision) the membrane fractionation of Ykt6-3E (n=4) in the novel Figure 3D and 3E, which has similar levels of Calnexin in Ykt6-WT and Ykt6-3E.
3) There may be a typo in line 332. please revised the sentence.
Thank you, now corrected.
Reviewer 2 Report
The paper has now been considerably improved.
Author Response
Thank you for the positive evaluation.
This manuscript is a resubmission of an earlier submission. The following is a list of the peer review reports and author responses from that submission.
Round 1
Reviewer 1 Report
The study addresses an important topic on the regulation of a critical SNARE protein related to autophagy, ER-Golgi trafficking, and secretion. There is little known about the regulation of ykt6 by post-translational modifications other than lipidation, and this study addresses the important topic on how phosphorylation could be influencing its function. There are some interesting findings, but enthusiasm is hampered by several technical issues, internal inconsistencies and contradictions throughout the work. Specific comments are below.
1) The introduction is well written but should be expanded to include key findings on the putative functions of ykt6 including ER-Golgi trafficking (McNew et al JBC 1997), a role in autophagy (Nair et al, Cell, 2011) and lysosomal stress (Cuddy et al, Neuron, 2019).
2) Figure 2A requires quantification to demonstrate the percentage of cells that show perinuclear location of ykt6-3E. Colocalization with organelle markers would help to show where the 3E mutant is going (perhaps Golgi).
3) The expression of each ykt6 construct is variable throughout the manuscript. The total levels are important to match, since overexpression alone may artificially drive ykt6 into membrane fractions or cause carry over in the fractionation procedure. For example, WT ykt6 in figure 2B is much lower compared to some 3E mutants, and 195 appears much lower overall. Line 176 states that WT and 3E are detected at similar levels in cytoplasm, but 6-fold higher in membrane, indicating that total (cytoplasmic + membrane ) 3E is expressed and detected at much higher levels than WT. If the effect of 3E mutant is to drive the protein into membranes, then the cytoplasmic fraction should be decreased in 3E.
4) The blot in figure 2D indicates some technical issue with the fractionation procedure. Calnexin is barely detectable in the supplemental figure 2D, but it is exposed longer in the main figure 2D. The supplement and main figure exposures should be matched. The calnexin blot in the main figure indicates that membrane contamination occurs in the cytosolic fraction, thus questioning the integrity of the assay.
5) It is not clear how biotin labeling efficiency indicates open / closed conformations. There are other more direct assays that could be done to address this question. There are many other things that could control labeling efficiency besides conformation.
6) There are inconsistencies between the assays that examine the distribution and conformation of ykt6 mutants. The blots in figure 2B show that ykt6-3E is about 50% cytosolic and 50% membrane, while panel F shows it at 22%. Similarly, 195 is not present in the membrane fraction in the blots, but 2F indicates that it is 11%, which is not much different from F42A at 14%, a mutant that has been well-established in the literature to drive ykt6 into membranes. The text classifies the 195 mutant as completely cytosolic (line 120), but then shows 195 mutant at 11% membrane bound, similar to the 3E/194/195 mutant, which they state are localizing to the membrane (line 150). Another recent paper also showed that C195S ykt6 could increase membrane localization compared to WT (Cuddy et al, Neuron, 2019), and this date seem to be consistent with these findings. Terminology should be consistent throughout, since as it is written now, it is confusing. Further, statistical analysis should be done for panel 2F. Please provide some explanation for the discrepancies between these assays.
7) If there is 0-1% of ykt6 WT in membrane fractions (figures 2B, D, and F), it is difficult to imagine how proteomic results of interacting proteins could have been obtained. Please provide some explanation for this.
8) Figure 4 requires quantification of the imaging.
9) It is unclear why a constitutively active Rab5 was used in figure 5, since this will alter the rate of endocytosis. The experiment should be repeated by labeling endosomes with endogenous markers, and without overexpressing artificial constructs that change the physiology of the cell. It is also not clear from the images how vesicle size could be obtained, since the resolution is not adequate – all of the signal appears to be coalesced into one region. A zoom image of a single vesicle would help to determine how the quantification in panel B was made.
10) There is no loading control shown for figure 5C. The effect of knock-down and overexpression of ykt6 on EV formation is confusing and requires some further explanation and more experiments to confirm these findings. In the EV assay, it is not clear if cell number or division/ cell death has been taken into account for normalization. If Ykt6 functions to promote EV secretion, the overexpression of WT should increase particles in panel H. Perhaps another functional assay should be used to measure secretion, or probe other functions such as autophagy or ER-Golgi trafficking assays.
11) If 3E promotes membrane localization, and membrane location will activate the protein’s function (line 286), why doesn’t 3E increase ykt6 secretory function or rescue the lethality phenotype in Drosophila? This is a confusing issue that should be resolved prior to publication.
12) Performing co-IPs for SNARE complexes may help to elucidate the connection between membrane location and function.
13) Reference #35 is incomplete.
Reviewer 2 Report
This paper summarizes data on a detailed study demonstrating that phosphorylation can be a major determinant defining the transition of Ykt6, a SNARE protein lacking a transmembrane domain, from soluble to membrane-associated forms. These membrane-associated states would mediate the regulation by Ykt6 of several important cellular processes. The topic is highly relevant and the conclusions and the perspectives open by this work on the mechanism of action of SNARES such as Ykt6 can be of high general interest.
The comparison of the behavior “in cell” of different forms of Ykt6, including its wild type sequence and mutants with supposed phosphorylation sites modified convincingly demonstrate that phosphorylation may mediate conformational changes that are required for the protein to properly expose membrane-interaction motifs. The experiments seem to have been carried out carefully, and presented in a clear manner in detailed figures.
The discussion of the paper is a bit short, and the paper could perhaps benefit of some additional discussion with respect to different questions:
- It is mentioned that there are other SNARES regulated by phosphorylation. However, in most of those cases phosphorylation prevents membrane association and SNARE-mediated membrane actions. This is opposite to the effect seen in Ykt6, in spite that phosphorylation sites in Ykt6 have been identified by homology analysis with other SNARES. The discussion should explain explicitly how phosphorylation in homologous residues is producing opposite effects in Ykt6 and other SNARES.
- Also, additional discussion should explain and recapitulate in a clearer way why the non-phosphorylable version of Ykt6 mimics some features and effects of the WT protein (i.e., rescue Ykt6 knockdown phenotypes) in spite that it is not in principle able to associate with membranes. Perhaps a summary of activities played by the non-phosphorylated and the phosphorylated forms could help the readers to end with a perspective of how much this work is advancing on the comprehension of the role of Ykt6 in cell biology.
- The existence of different potential phosphorylation sites in Ykt6 suggests the possibility that different phosphorylation levels, perhaps as a consequence of the action of different kinases, could play differential roles. Also, the possibility that one of those sites could be the critical feature defining the switch between soluble and membrane bound forms. Perhaps the authors could like adding some discussion in this respect.
- A list of abbreviations could aid the readers to follow the text.